# An integrated model for predicting KRAS dependency

Yihsuan S. Tsai[1,2], Yogitha S. Chareddy[1], Brandon A. Price[1,2], Joel S. Parker[1,2], Chad V. Pecot [1,3]*

**1** UNC Lineberger Comprehensive Cancer Center, University of North Carolina at Chapel Hill, Chapel Hill, North Carolina, United States of America, **2** Department of Genetics, University of North Carolina at Chapel Hill, Chapel Hill, North Carolina, United States of America, **3** Division of Hematology & Oncology, University of North Carolina at Chapel Hill, Chapel Hill, North Carolina, United States of America

☯ These authors contributed equally to this work.
* pecot@email.unc.edu

**Data Availability Statement:** All of the data are publicly available as part of the DEMETER2 and CCLE datasets. DEMETER2 version 5 can be found here: https://figshare.com/articles/dataset/DEMETER2_data/6025238/5. CCLE data can be

## Abstract

The clinical approvals of KRAS G12C inhibitors have been a revolutionary advance in precision oncology, but response rates are often modest. To improve patient selection, we developed an integrated model to predict KRAS dependency. By integrating molecular profiles of a large panel of cell lines from the DEMETER2 dataset, we built a binary classifier to predict a tumor's KRAS dependency. Monte Carlo cross validation via ElasticNet within the training set was used to compare model performance and to tune parameters α and λ. The final model was then applied to the validation set. We validated the model with genetic depletion assays and an external dataset of lung cancer cells treated with a G12C inhibitor. We then applied the model to several Cancer Genome Atlas (TCGA) datasets. The final "K20" model contains 20 features, including expression of 19 genes and KRAS mutation status. In the validation cohort, K20 had an AUC of 0.94 and accurately predicted KRAS dependency in both mutant and KRAS wild-type cell lines following genetic depletion. It was also highly predictive across an external dataset of lung cancer lines treated with KRAS G12C inhibition. When applied to TCGA datasets, specific subpopulations such as the invasive subtype in colorectal cancer and copy number high pancreatic adenocarcinoma were predicted to have higher KRAS dependency. The K20 model has simple yet robust predictive capabilities that may provide a useful tool to select patients with KRAS mutant tumors that are most likely to respond to direct KRAS inhibitors.

## Author summary

Mutant KRAS drives approximately 25% of all cancers and has traditionally been considered "undruggable". However, the recent clinical approvals of inhibitors targeting KRAS with the specific G12C mutation in lung cancer has shepherded in a new era in precision medicine. Although promising, the responses are often modest and short-lived. Therefore, the ability to predict which tumors are dependent on KRAS will help select patients most likely to derive clinical benefit, and those who will not. We have developed an integrated

found here (from the Depmap 18Q1 version): https://depmap.org/portal/download/all/.

**Funding:** Y.C. was supported in part by funding from the National Institute of General Medical Sciences (https://www.nigms.nih.gov/) of the National Institutes of Health (NIH) under the Program in Translational Medicine T32 (award number GM122741). C.V.P. was supported in part by the NIH (https://www.nih.gov/) (award numbers R01CA215075, R01CA258451 and 1R41CA246848), the Lung Cancer Research Foundation, the Free to Breathe Metastasis Research Award and a North Carolina Biotechnology Translation Research Grant (NCBC TRG). The funders had no role in study design, data collection and analysis, decision to publish, or preparation of the manuscript.

**Competing interests:** We have read the journal's policy and the authors of this manuscript have the following competing interests: C.V.P. is the founder of EnFuego Therapeutics, Inc, which is focused on the development of KRAS therapeutics and holds equity in the company. The remaining authors disclose no potential conflicts of interest.

"K20" model based on features that can improve prediction of KRAS-dependency beyond the presence of an activating KRAS mutation. When applied to lung adenocarcinoma, pancreatic adenocarcinoma, and colorectal cancer patient datasets, the K20 model identified specific subpopulations that correlate with greater dependency on KRAS. These findings present a novel approach for identifying biomarkers that can aid in the selection of patients who are most likely to benefit from KRAS inhibitors.

## Introduction

Dysregulation of the RAS family of GTPases is responsible for driving nearly 30% of all cancer types (Catalogue of Somatic Mutations in Cancer [COSMIC] v92). Discovered nearly four decades ago as the oncogenes NRAS, KRAS, and HRAS, the RAS family has expanded to include approximately 150 members involved in important cellular processes like cell division, differentiation, migration, and apoptosis [1]. In healthy cells, membrane-bound RAS-family proteins remain inactive while bound to GDP until stimulated by extracellular signals, which will cause the formation of an intermediate complex with GTP and initiate several downstream signaling cascades. However, missense mutations in RAS proteins render them constitutively active in the GTP-bound state and have been demonstrated to promote nearly all "hallmarks of cancer" [2,3].

Of the three major RAS-family isoforms, mutated KRAS comprises 84% of all RAS-driven diseases and propagates many aggressive tumor types, including lung, colorectal, and pancreatic cancer [4]. The role of KRAS in cancer progression is well-studied, but until recently the protein has largely been considered "undruggable". Due to the structure and surface topology of the GTPase, traditional small-molecule inhibitors that can directly antagonize the protein's function have been widely regarded as untenable. Attempts to block factors involved in MAP kinase signaling (such as MEK and Raf) or binding partners to KRAS have shown some clinical promise in specific cancer types but have presented challenges with toxicity and eventual treatment resistance even in combination with other treatment options [5–7].

Nearly all KRAS mutations are concentrated at three codons: glycine-12 (G12), glycine-13 (G13), and glutamine-61 (Q61) [4]. Recent advances in medicinal chemistry have identified a binding pocket in the glycine-to-cysteine missense mutant KRAS protein at amino acid 12 (G12C), which comprises approximately 12% of all KRAS cancer mutations and accounts for a substantial fraction of KRAS mutations in non-small cell lung cancer (13%), and to a lesser degree in colorectal (4%) and pancreatic cancers (1–3%) [4,8–11]. Tool compounds developed by Ostrem and colleagues covalently bind to the mutant cysteine and extend into the binding pocket primarily containing the switch II region (S-IIP), generating a selective response in mutant cells by repressing signaling [12]. Similar compounds discovered by additional groups have led to the rapid development of direct KRAS G12C inhibitors, the most clinically advanced of which are MRTX849 (adagrasib) and AMG510 (sotorasib) [13–15], both of which are now FDA-approved for use in lung cancer.

Importantly, previous work has revealed that not all KRAS-mutant cell lines are KRAS-dependent, and that such KRAS-independent cancers exhibit features of an epithelial-mesenchymal transition (EMT) and apoptosis resistance [16]. De novo resistance to blockade of KRAS-activated MAP kinase signaling has recently been linked with high expression of the EMT regulator zinc finger E-box binding homeobox 1 (ZEB1) [17]. Additionally, genetic models of KRAS depletion have revealed rapid development of several cell intrinsic and non-cell autonomous mechanisms of KRAS-independence [18–20]. These data imply that direct

inhibitors of mutant KRAS may not be as efficacious as is seen with other "oncogene-addicted" cancers, and numerous mechanisms of resistance may emerge. Consistent with this, while response rates of ~60–70% are typically seen in lung adenocarcinomas (LUAD) using inhibitors of aberrant EGFR, ALK, ROS1 or RET, the confirmed response rate in the expansion dose (960 mg) cohort for AMG510 in LUAD patients was about 35%. Even lower, the expansion cohort of colorectal cancer (CORE) patients had a response rate of 12% [15]. Taken together, with the emergence of well-tolerated and potent KRAS G12C inhibitors, there is an urgent need for a biomarker signature that can select patients most likely to benefit from these treatments beyond the presence of an activating KRAS mutation. While previous groups have developed models to determine RAS activation and dependency in specific cancer types, there is no model that has identified a gene signature across all cancer types with predictive capabilities of KRAS dependency beyond the RAS pathway [16,21,22]. By integrating several publicly available datasets that couple KRAS dependencies, genomic features, and tumor-specific transcriptional profiles, we have developed an integrated model (henceforth the K20 model) that improves prediction of KRAS dependency beyond the presence of an activating KRAS mutation.

## Results

### An integrated dataset for modeling KRAS dependency

We used DEMETER2 as our KRAS cancer dependency dataset as it combined three large-scale RNAi screen datasets and integrated them with model-based normalization [23]. This model system estimates gene dependency on an absolute scale with a score of zero representing no dependency, higher/positive scores representing resistance, and lower/negative scores representing sensitivity (i.e. impaired cell growth upon gene silencing). This dataset allowed us to use RNAi screen datasets from Project Achilles [24], Project DRIVE [25] and a smaller breast cell line cohort [26] and integrate it with publicly-available multi-omic datasets from the Cancer Cell Line Encyclopedia (CCLE) [27]. There were 712 cell lines in total from DEMETER2. As expected, the KRAS dependency scores, which represent absolute KRAS dependency, were significantly lower in KRAS-mutant cells compared to KRAS wild-type (WT) cell lines (Panels A and B in S1 Fig). Using K-means clustering with k = 3, the 712 cell lines were divided into sensitive (n = 29), intermediate (n = 112) and refractory (n = 571) clusters (Panel C in S1 Fig). Within the 126 KRAS-mutant cell lines, there was enrichment of sensitive and intermediate cell lines compared to KRAS wild-type cell lines, which were more refractory (Fisher-exact p-value = 6.2e-55, Panel D in S1 Fig). Among the 126 KRAS-mutant cell lines, lung, pancreas and colorectal were the top three diseases with the highest representation (Panel E in S1 Fig). Because the cell types and expression profiles of the hematologic (e.g. leukemia, multiple myeloma and lymphoma) and non-carcinoma solid tumor cell lines (e.g. CNS tumors) were very different from carcinoma cell lines, we excluded those and cell lines of rare diseases from further analysis. In total, there were 444 carcinoma cell lines with expression, exon mutation and dependency data, of which 106 had KRAS mutations. Fig 1A shows all of the 444 cell lines sorted by KRAS dependency score with color bars to indicate activating mutations in KRAS and other genes in the Ras/Raf pathway.

### Development of a KRAS dependency classifier

To model "clinical benefit", which we deemed reflective of tumors either shrinking or becoming cytostatic upon direct KRAS targeting, we grouped cell lines in the "sensitive" (n = 24) and "intermediate" (n = 83) clusters into a "non-refractory" class. We aimed to build a prediction model that would classify cell lines into binary groups (non-refractory vs. refractory). The 444

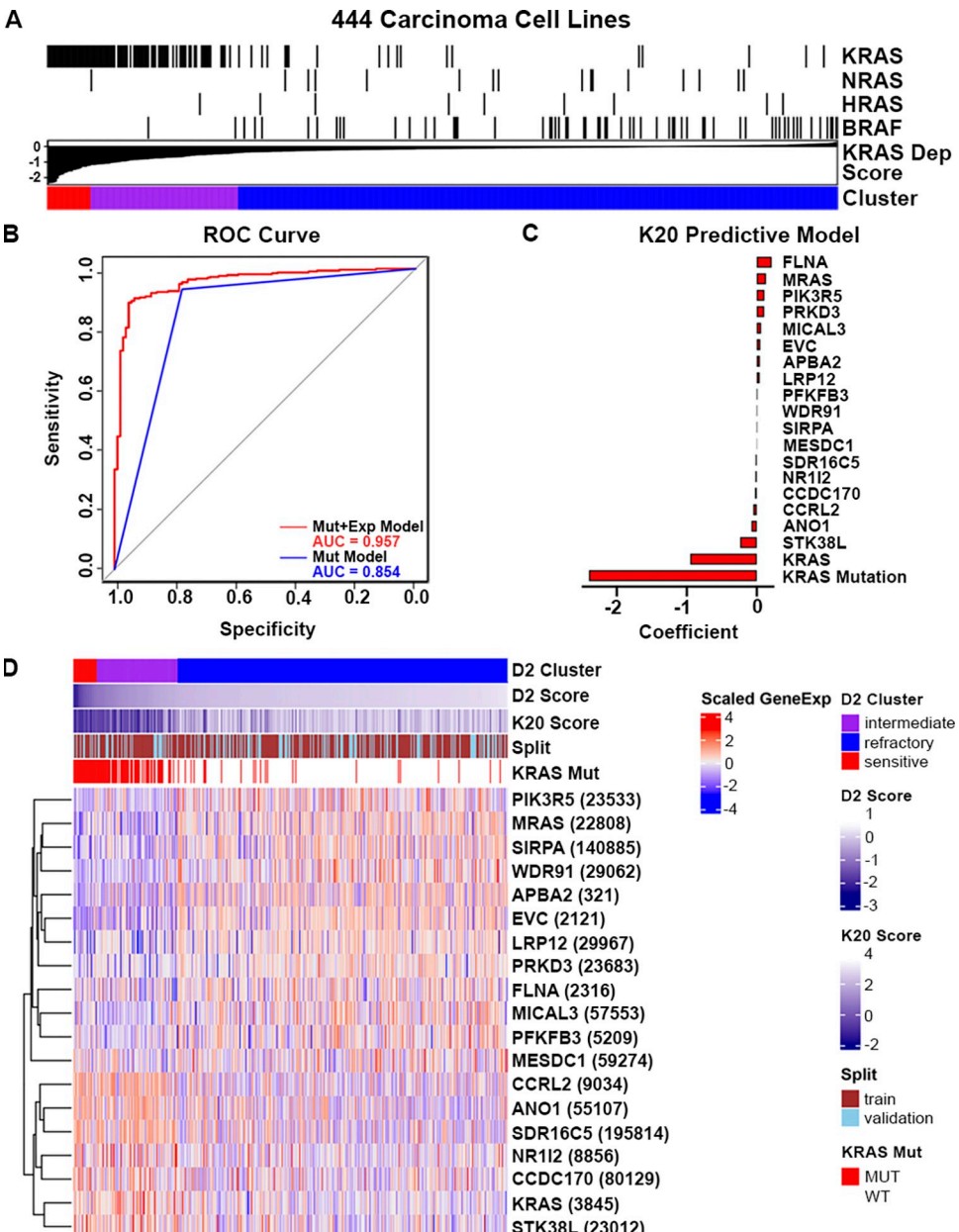

**Fig 1. KRAS dependency in solid cancer cell lines and performance of the K20 model.** Cell lines in the training (N = 298) and validation (N = 146) sets had consensus with DEMETER2 KRAS dependency scores. The 20 predictive features in the model were identified from CCLE gene expression and mutation data. (A) Waterfall plot showing KRAS dependency scores of all carcinoma cancer cell lines from DEMETER2 with color bars indicating cell lines with activating mutations in KRAS, NRAS, HRAS, and BRAF. (B) Receiver operating characteristic (ROC) curve comparing the K20 model performance (in red) with the KRAS mutation only model (in blue). (C) Bar plot of K20 model coefficients. (D) Heatmap showing gene expression of 19 classifier genes and mutation status of KRAS in the combined training (in brown) and validation sets (in sky blue) for all 444 cell lines. Entrez ID is indicated in parentheses next to each gene. Red represents higher expression and blue represents lower expression.

solid cancer cell lines were randomly divided into training (67%, 72 KRAS-mutant and 226 KRAS WT) and validation (33%, 34 KRAS-mutant and 112 KRAS WT) sets, balancing for tissue types and KRAS dependency clusters. Monte Carlo cross validation via ElasticNet [28] was used to develop and compare the prediction accuracy of KRAS dependency among different

features in the training set and to pick the best tuning parameters. We compared five different feature sets to see which one had better KRAS dependency prediction: [1] gene expression only, [2] gene expression and gene level mutation status (binary), [3] gene expression and KRAS mutation status (binary), [4] gene expression and specific KRAS mutation type and [5] gene expression, KRAS mutation status and the interaction between them (Panel A in S2 Fig). For gene expression, in order to select genes that are differentially expressed in tumors and avoid genes that are cell-line specific, we only included genes whose expression had variations (standard deviation >0.5 in log2 normalized gene expression) in the lung adenocarcinoma (LUAD), pancreatic adenocarcinoma (PDAC), and colorectal cancer (CORE) mRNA datasets from The Cancer Genome Atlas (TCGA) [22,29–31]. The feature set 5 with "KRAS mutation interaction" had the highest AUC at 0.94 (shown as a black dotted line in S2A Fig) when setting the α at 0.3 and the number of features at 80, while the feature set 3 with "gene expression and KRAS mutation status" had the second highest AUC at 0.93 when using 20 features at α = 0.9 (Panel A in S2 Fig). We selected the feature set with "gene expression and KRAS mutation status" as our final model because of the minimal performance difference (i.e., AUC = 0.94 vs. 0.93) and the smaller number of features required. We named this final model as "K20", which includes the gene expression of 19 genes and the mutation status of KRAS. Panel B in S2 Fig shows the AUC comparison among all feature sets with a small number of features. Feature set 4 ("gene expression with specific KRAS mutation type") performed similarly to feature set 3 and had only subtle feature differences. When applying the K20 model to the whole dataset, it had an AUC at 0.96, which is an improvement upon the KRAS mutation status only model (AUC = 0.85) (Fig 1B). The performance improvement was seen in the validation set as well (Panel C in S2 Fig). Fig 1C shows the weight of all 20 features included in the final model. While KRAS mutation status is a binary variable (KRAS wild-type = 0, KRAS-mutant = 1), the rest of the features are based on gene expression (RPKM on log2 scale). Importantly, aside from KRAS mutational status, we found that KRAS expression itself played the strongest role in predicting KRAS dependency (Fig 1C). Indeed, for KRAS WT cell lines, higher KRAS expression was significantly associated with increased KRAS dependency, and we see a similar trend for KRAS-mutant cell lines (Panel C in S3 Fig). High KRAS expression is oncogenic by nature [32] and generally correlates with poor prognosis, and some clinical studies have assessed the differential impact of KRAS expression over KRAS mutation in certain cancers. Increased KRAS amplification, as opposed to KRAS mutations, can lead to increased metastatic endometrial disease [33]. In colorectal cancer, independent of the presence of KRAS mutations, KRAS copy number gain and amplification can also be a negative predictor to anti-EGFR treatment [34,35]. Previous modeling of KRAS dependency found that KRAS gene copy number and protein expression were correlated with an increased dependency on the oncogene [16]. Fig 1D shows a heatmap of all 20 features in the K20 prediction model. Cell lines were sorted by DEMETER2 KRAS dependency scores. We found that higher expression in CCRL2, NR1l2, CCDC170, ANO1, SDR16C5 and KRAS occur more frequently in KRAS-dependent cell lines, while expression in genes such as SIRPA, MRAS, LRP12, EVC and APBA2 are higher in KRAS-independent cell lines. Several genes, such as STK38L and SDR16C5, were previously shown to correlate with oncogenic KRAS [36,37], and CCRL2 expression on cells within the TME is known to suppress KRAS-mediated tumor progression [38].

## External validation of the K20 model

Upon sorting the cell lines by K20 prediction scores, which represent predicted KRAS dependency based on our model, we selected a prediction score of 1.477 as a cutoff with the maximal

sum of sensitivity plus specificity. Fig 2A shows the original DEMETER score and Fig 2B shows the samples sorted by K20 prediction scores. Except for five cell lines, all sensitive and intermediate cell lines were predicted to be non-refractory, giving an AUC score of 0.94 in the validation set (Panel B in S2 Fig). The model prediction scores are highly correlated with the DEMETER2 KRAS dependency scores (Fig 2C), showing that cell lines with low prediction scores are sensitive to KRAS depletion and cell lines with high prediction scores are resistant to KRAS depletion, while cell lines with prediction scores in between are likely to have a partial response. To validate the predictive performance of the K20 model, we first performed *in vitro* RNAi-mediated dose-response assays and assessed comparative cell viability following KRAS knockdown. We selected several cell lines that were derived from LUAD, PDAC, or CORE tumors, of which one cell line was KRAS WT and the other cell lines represented unique KRAS mutations in codons 12 and 13. Based on our K20 model, MIA-PaCa-2 (G12C mutant pancreatic cancer), H441 (G12V mutant lung cancer), and SK-CO-1 (G12V mutant colon/ colorectal cancer) were selected from the sensitive cluster, A427 (G12D mutant lung cancer), HCT116 (G13D mutant colorectal cancer), H727 (G12V mutant lung cancer), and KE39 (KRAS WT gastric adenocarcinoma) were selected from the intermediate cluster, and NCI-H2030 (H2030) (G12C mutant lung cancer) and NCI-H1355 (H1355) (G13C mutant lung cancer) were selected from the refractory cluster. The cell lines were transfected using a 10-point concentration gradient with two different pan-KRAS siRNAs that were previously characterized to show potent KRAS knockdown (Panel A in S3 Fig) [39]. Using metabolic activity as a readout for cell viability, we found that each KRAS siRNA produced similar data that followed the general clustering trend predicted by the K20 model (Fig 2D, Panel A in S3 Fig). For example, MIA-PaCa2 and SK-CO-1 produced low GI50s at 0.94 and 1.76 nM, respectively, upon treatment with KRAS siRNA #1, indicating a higher sensitivity to KRAS knockdown, whereas H2030 and H1355 were about 5 to 7-fold more refractory to KRAS silencing with GI50s at 5.39 and 6.67 nM, respectively. Interestingly, KE39 was shown to be quite sensitive to KRAS knockdown despite being KRAS WT. Our K20 prediction score for this line is largely driven by its high KRAS expression levels. Together, these findings show that the K20 model can predict a range of KRAS dependencies across several KRAS mutations, various cancer subtypes, and even dependencies of cancers that are KRAS WT. To evaluate an external dataset, we compared K20 prediction scores with GI50s from a panel of KRAS G12C-mutant lung cancer cell lines treated with the G12C inhibitor, ARS1620 [40]. This dataset was particularly useful since it included 12 cell lines with a large dynamic range of KRAS dependencies. We found that the K20 prediction scores were highly correlated with the GI50 readout (Fig 2E), further supporting the predictive capabilities of the K20 model.

## Molecular features of predicted KRAS-Dependency in human tumors

We then applied the K20 model to TCGA-PANCAN. As expected, we found that patients with KRAS mutations from all cancer types had significantly lower prediction scores, indicating higher KRAS dependency (Fig 3A), which was also seen in all CCLE tissue types (Panel D in S3 Fig). In all cancer patients with KRAS mutations, we found gastrointestinal (GI) tract cancers: esophageal carcinoma (ESCA), stomach adenocarcinoma (STAD), colon adenocarcinoma (COAD), rectum adenocarcinoma (READ) and pancreatic adenocarcinoma (PAAD), had the lowest K20 prediction scores. Interestingly, many KRAS wild-type GI tract cancer patients also had low K20 prediction scores, which we suspect may be due to false negatives in KRAS mutation detection or the presence of mutations in genes like NRAS or BRAF which may phenocopy oncogenic KRAS. Melanoma patients were predicted to be the most resistant cancer type to KRAS dependency, regardless of KRAS mutation status. Fig 3B shows a

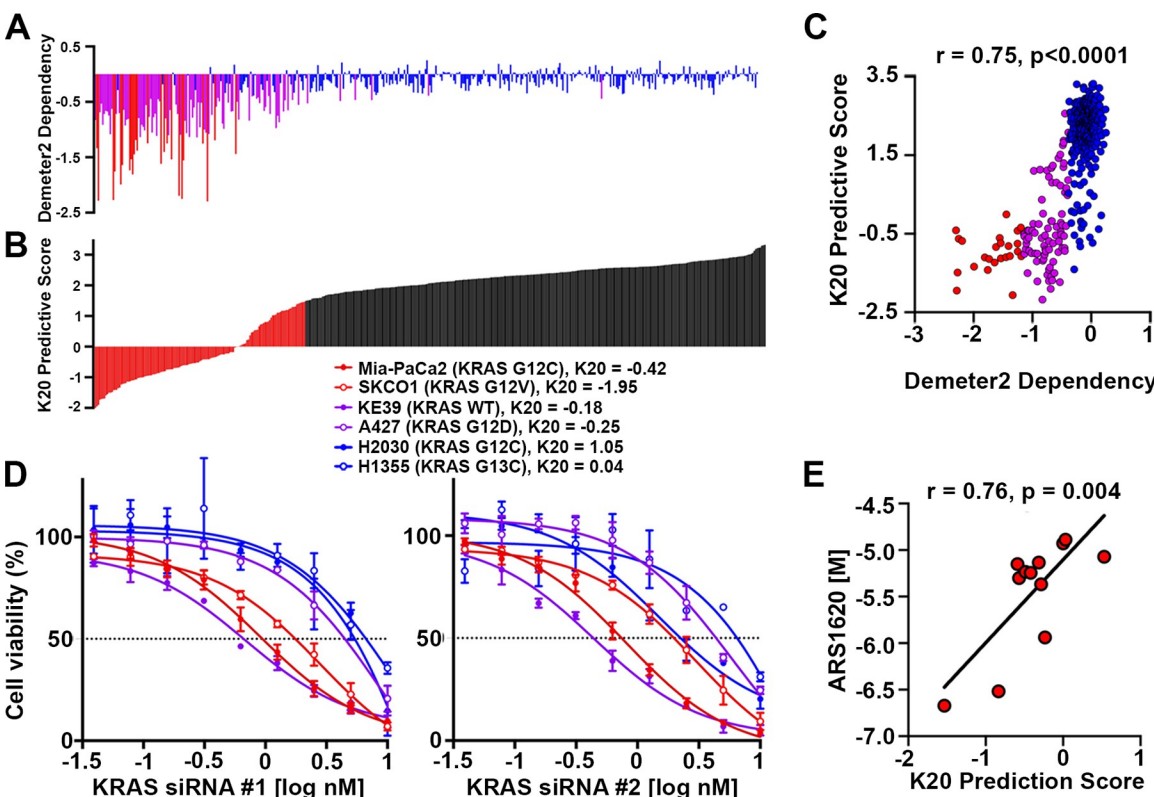

**Fig 2. Independent validation of K20 model.** Validation of K20 model was conducted in internal validation set and two external sets. Waterfall plots of KRAS dependency scores from DEMETER2 in (A) and of K20 prediction scores in (B). Cell lines were sorted from the lowest K20 prediction scores to the highest prediction scores from left to right (red: score < threshold, black score ≥ threshold). (C) Scatter plot of KRAS DEMETER2 scores (in x-axis) vs. K20 prediction scores (in y-axis) in all cell lines shows high correction (Pearson r = 0.75, p<0.0001). (D) Representative dose-response curves for six different cancer cell lines treated for six days with two KRAS-targeting siRNAs. CellTiter Glo was used to measure cell viability. Fitted curves generated by GraphPad Prism. (E) Scatter plot of K20 prediction scores (in x-axis) vs. GI50 scores of ARS1620 (in y-axis) in 12 lung cancer cell lines shows high correlation (Pearson r = 0.76, p = 0.004) in a public dataset. Red, purple and blue represent sensitive, intermediate and refractory, respectively in A, C, and D.

heatmap of the gene expression of the 19 genes in K20 model and KRAS mutation status for each patient sorted by K20 prediction scores.

It is possible that gene expression levels in cell lines cannot capture changes caused by the tumor microenvironment. Therefore, we used the AJIVE (Angle-based Joint and Individual Variation Explained) method [41] to measure the joint variation shared between tissues (i.e. TCGA) and cell lines (i.e. CCLE) and calculated the CCLE joint statistics [42]. Genes with higher CCLE joint statistics represent improved translation from cell line to TCGA data. All 19 genes in the K20 model passed a 5% FDR threshold, showing increased joint-behavior with TCGA over background (Figs 3C and S4), which indicates that the signature may translate well to actual tumor biopsies.

Because LUAD, CORE and PDAC have some of the highest KRAS mutation frequencies of all solid tumors, we specifically tested the association between their K20 prediction scores and tumor characteristics. In CORE cancers, we found that 3 out of the 93 tumors with KRAS mutations were predicted to be resistant to KRAS depletion (Panel A in S5 Fig). These patient tumors had lower KRAS gene expression in comparison to KRAS-mutant tumors that were predicted to be sensitive (p = 0.0035). To see if molecular subtypes had differential KRAS dependencies, the prediction scores were evaluated across TCGA mRNA expression subtypes,

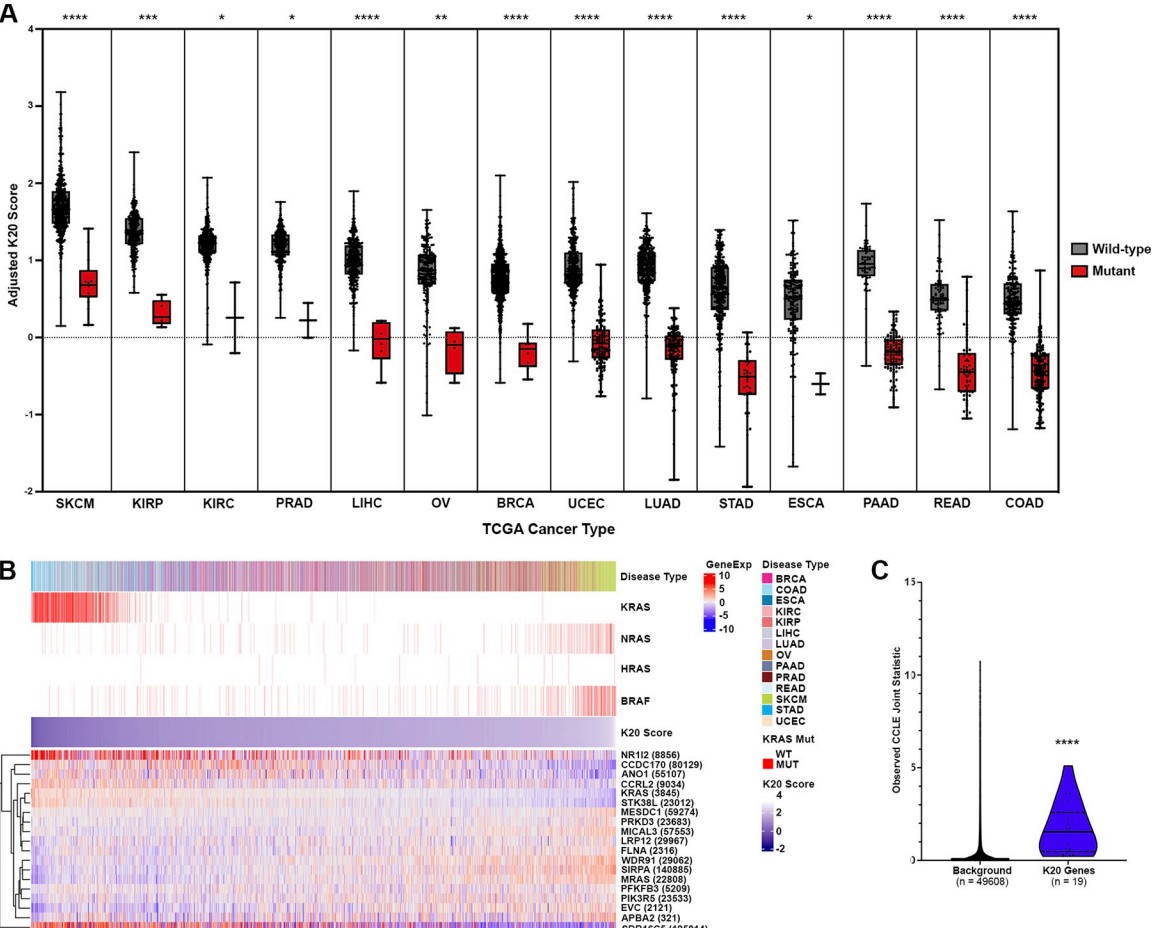

**Fig 3. Applying the K20 predictive model to TCGA-PANCAN (14 cancer types).** (A) Box plot of K20 prediction scores in 14 TCGA solid tumor types separated by tissue type (x-axis) and KRAS mutation status. Each sample is represented as a dot. (B) Heatmap of the gene expression of the 19 model genes. Disease type is shown in the top color bar. KRAS, NRAS, HRAS, and BRAF mutation statuses are shown in the second, third, fourth, and fifth color bars respectively. K20 model prediction scores are shown in the sixth color bar. (C) Box plots of CCLE joint statistics in K20 model genes versus background genes. Wilcoxon rank-sum test was used for comparisons. (****) = p<0.0001, (***) = p<0.001, (**) = p<0.01, (*) = p<0.05. BRCA: Breast invasive carcinoma, COAD: Colon adenocarcinoma, ESCA: Esophageal carcinoma, KIRC: Kidney renal clear cell carcinoma, KIRP: Kidney renal papillary cell carcinoma, LIHC: Liver hepatocellular carcinoma, LUAD: Lung adenocarcinoma, OV: Ovarian serous cystadenocarcinoma, PAAD: Pancreatic adenocarcinoma, PRAD: Prostate adenocarcinoma, READ: Rectum adenocarcinoma, SKCM: Skin Cutaneous Melanoma, STAD: Stomach adenocarcinoma, UCEC: Uterine Corpus Endometrial Carcinoma.

methylation subtypes, MSI status, and hypermutation status [30]. We found patients in the invasive gene expression subtype were predicted to be more sensitive to KRAS loss regardless of their KRAS mutation status (Panel B in S5 Fig). Additionally, we also compare the K20 prediction scores with the Consensus Molecular Subtypes from Guinney et al. [22]. We found that patients with the CMS4, or mesenchymal, subtype were predicted to have more refractory tumors, while CMS3, or metabolic, subtype tumors were predicted to be more sensitive (Panel F in S5 Fig).

In PDAC patients, almost all patients had KRAS alterations (90.7%), but the prediction scores fell within a large range (-10 to -1) (Fig 4A). Except for one tumor, all tumors with KRAS mutations had lower K20 prediction scores than KRAS WT tumors (Fig 4A). Tumors in the progenitor subtype [31], or those with copy number aberrations, were predicted to be the most sensitive to KRAS inhibition (Fig 4B and 4C). While nearly all tumors were predicted

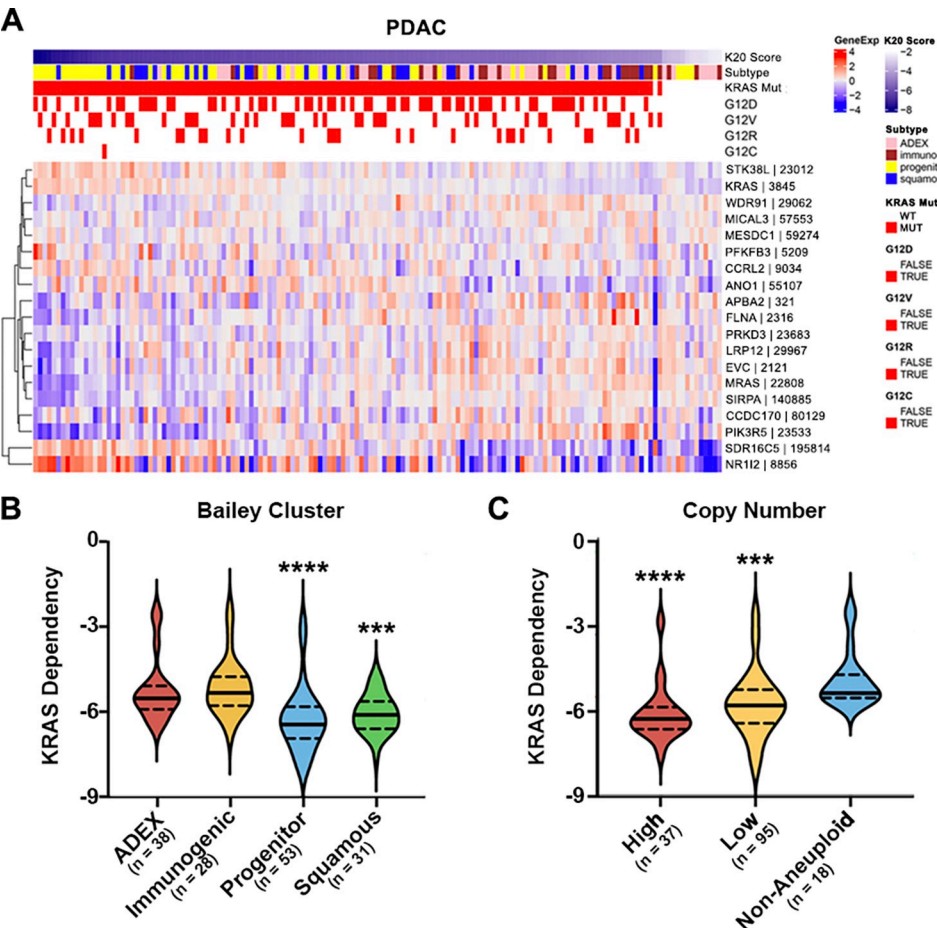

**Fig 4. Apply K20 predictive model to TCGA pancreatic adenocarcinoma (PDAC) patients.** The K20 model predicts that most of the pancreatic adenocarcinoma patients with KRAS mutation are not KRAS-refractory. The model prediction scores are associated with Bailey clusters and copy number. (A) Heatmap showing gene expression of 19 classifier genes and KRAS mutation status and types in TCGA pancreatic adenocarcinoma patients. Samples were sorted by prediction scores (left to right: lowest to highest). (B-C) Violin plots of K20 prediction scores by (B) Bailey cluster (progenitor vs ADEX, p = <0.0001; progenitor vs immunogenic, p = <0.0001; squamous vs ADEX, p = 0.0002; squamous vs immunogenic, p<0.0001), and (C) copy number (low vs non-aneuploid, p = 0.0002; high vs non-aneuploid, p<0.0001; low vs high, p = 0.0122). Wilcoxon rank-sum test was used for comparisons. (****) = p<0.0001, (***) = p<0.001.

to be highly dependent on KRAS, tumors in the progenitor and squamous subtypes had significantly lower dependency scores compared to tumors in immunogenic or aberrantly differentiated endocrine exocrine (ADEX) subtypes (Fig 4B). Interestingly, the ADEX subtype is typically associated with KRAS activation, whereas the quasi-mesenchymal squamous cluster is driven by p53 and KDM6A mutations [43] and the progenitor tumors express genes found in early pancreatic development [31]. Pancreatic cancer patients with non-aneuploid copy number were predicted to be less sensitive to KRAS loss than patients with copy number aberration (Fig 4C). This may be because of the chromosomal instability introduced by aneuploidy, which can be driven by KRAS mutations and easily perturbed [44]. Because only a small number of the PDAC patients were KRAS WT, the prediction score differences seen in Bailey expression subtypes and copy number clusters were primarily driven by patients with KRAS mutations (Panels A+B in S6 Fig). We did not find K20 prediction score differences between Moffitt subtypes (i.e., Basal vs. Classical, Panel D in S6 Fig) [45]. The K20 model predicted

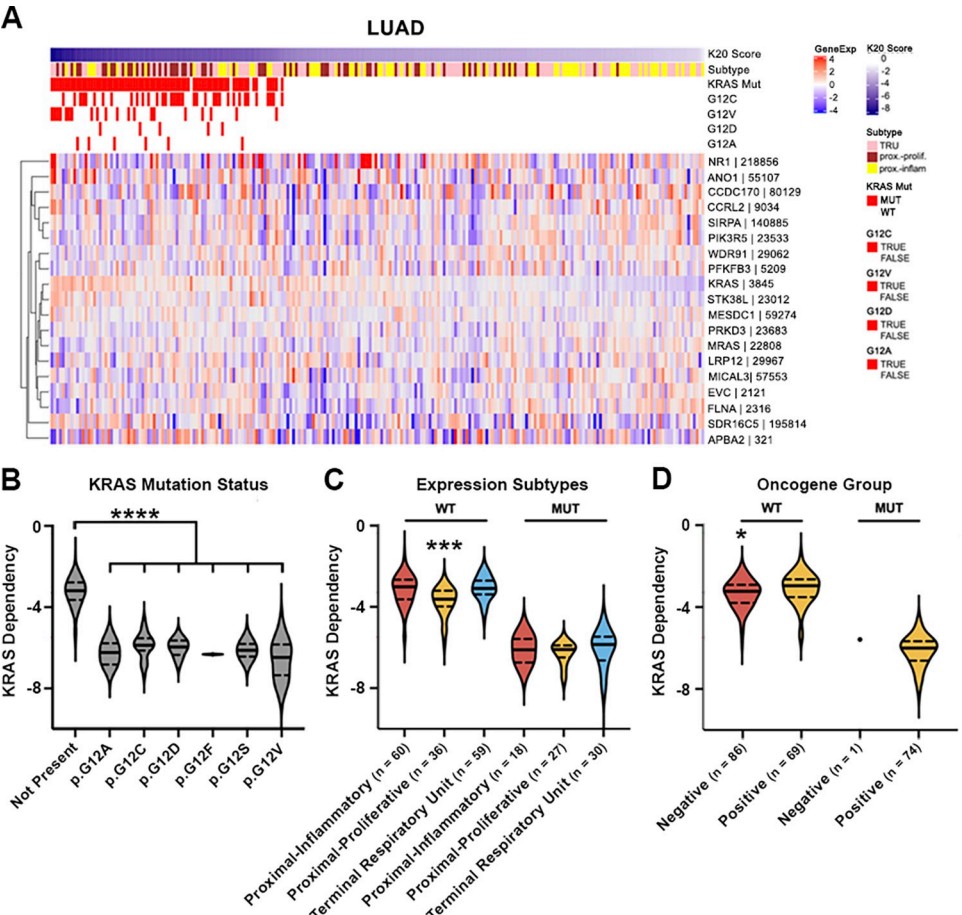

**Fig 5. Apply K20 predictive model to TCGA lung adenocarcinoma cancer (LUAD) patients.** K20 model predicts that most of the lung adenocarcinoma patients with KRAS mutation are KRAS-sensitive and some might be intermediate. The model prediction scores are associated with TCGA expression subtypes, STK11 mutation status, oncogene groups and transversion status. (A) Heatmap showing gene expression of 19 classifier genes and KRAS mutation status and types in TCGA lung adenocarcinoma patients. Samples were sorted by prediction scores (left to right: lowest to highest). (B-D) Violin plots of K20 prediction scores by (B) KRAS mutation types (not present vs present for any mutation, p<0.0001), (C) expression subtypes (KRAS-wildtype (WT): proximal-proliferative vs proximal-inflammatory, p = 0.0002; proximal-proliferative vs terminal respiratory unit, p<0.0001), and (D) oncogene groups (WT: negative vs positive, p = 0.0143). Wilcoxon rank-sum test was used for comparisons. (****) = p<0.0001, (***) = p<0.001, (*) = p<0.05.

scores were not associated with subtypes defined by lncRNA, methylation, RPPA or micro-RNA clusters (Panels C, E-G in S6 Fig).

In LUAD patients, almost all KRAS G12-mutant cancer patients were predicted to be KRAS dependent, albeit along a spectrum (Fig 5A and 5B), and we found that the prediction scores were only correlated with some TCGA subtypes in the KRAS WT tumors [29]. KRAS WT tumors in the proximal-proliferative expression subtype (which has a high incidence of STK11 mutations) were predicted to be more sensitive than the other two subtypes, and this trend was seen in KRAS-mutant tumors as well (Fig 5C). All the KRAS-mutant tumors were classified within the oncogene positive group, except one. About 45% of the KRAS WT tumors were oncogene positive, but they were predicted to be less sensitive to KRAS loss than KRAS WT tumors in the oncogene negative group (Fig 5D). This KRAS resistance in oncogene positive tumors without KRAS mutations is probably due to different driver mutations found in

those tumors, such as EGFR and BRAF mutations. In lung cancer, mutations in p53 and STK11 commonly co-occur with KRAS. However, our model did not predict KRAS dependency differences by the presence or absence of an STK11 mutation or p53 mutation when stratified by KRAS mutation status (Panels A+B in S7 Fig). More than 80% of KRAS-mutant tumors were in the transversion-high classification (Panel C in S7 Fig), while the transversion-low class were often enriched with EGFR-mutant tumors [29]. However, we did not find differences in prediction scores based on transversion classification within KRAS WT or mutant tumors (Panel C in S7 Fig).

## Discussion

Oncogenic drivers in cancer, such as KRAS, can be targeted by potent inhibitors and result in remarkable therapeutic benefit. However, defining oncogene dependency can be complex and context dependent. The heterogeneity of KRAS-driven disease across cancer types can obscure patterns of dependency. While mutant and hyperactive KRAS has been shown to induce and sustain cancer growth, other cellular programs can amplify these effects and provide alternative routes for cell survival upon KRAS inhibition. Using data from DEMETER2 and Project Achilles, we have developed a model with 20 features (the K20 model) to predict the KRAS dependency of carcinomas. While KRAS dependency may not be on a linear scale, we have defined three clusters of dependency that may help characterize and discriminate differences across various cancer subtypes and even genotypes.

The signaling networks of KRAS with other important pathways in cancer cells are well-documented, however the mechanisms of KRAS dependency remain poorly understood. The features within our K20 model offer a novel look into the genetic landscape of KRAS-dependent cancer cells. Some features have been directly or indirectly linked to KRAS in promoting tumor progression. For example, loss of the STK38L kinase, which can be co-amplified with KRAS, has been shown to result in decreased cell viability in KRAS-dependent pancreatic cells [36]. SDR16C5 has been found to be upregulated in KRAS-mutant rectal tumors compared to KRAS wild-type [37], and increased SIRPA may be indirectly linked to mutant KRAS through the overexpression of the EZH2 polycomb complex [46]. In contrast, ANO1 has been linked to MAPK signaling in a RAS-independent manner [47], and mutations in LRP12 have been linked with oncogenic KRAS [48]. Other features within our K20 model have also been shown to downregulate KRAS or compensate for its loss. For example, MICAL3 negatively correlates with KRAS dependency in our model and is frequently enriched in non-smoker related lung cancer [49]. Studies on PRKD3 have revealed its role in driving ERK/c-MYC-facilitated tumorigenesis in a KRAS-independent manner [50,51], suggesting a compensatory mechanism following loss of KRAS. Similarly, FLNA prevents KRAS-mutant lung metastasis and is often downregulated in KRAS-mutant tumors [52,53]. Evidence that PFKFB3 promotes tumorigenesis through hyperactivation of metabolism and can be targeted therapeutically to decrease the viability of KRAS-mutant pancreatic cells seems contradictory to the negative influence PFKFBB3 has on KRAS dependency in our model. However, this relationship may instead offer insight into a metabolic mechanism of resistance in KRAS-mutant cancers that is promoting survival independently of KRAS [54,55]. Future studies into specific features and related mechanisms within our model may further elucidate targetable vulnerabilities that can synergize with KRAS inhibitors and/or overcome mechanisms of resistance.

Our K20 predictive model was validated using several *in vitro* dose-response studies, which show clear differences in response, especially between the sensitive and refractory groups irrespective of mutation status. Our RNAi data with the lung H2030 cell line (KRAS G12C mutant) shows that identification of a KRAS mutation is not sufficient to justify treatment

with KRAS G12C inhibitors. Our external validation of the K20 model across a larger panel of lung KRAS G12C mutant cell lines accurately predicted the dependency of inhibition with ARS1620, which revealed a clear spectrum of KRAS dependency. These findings underscore the need for a predictive tool to *a priori* predict a tumor response to direct KRAS inhibition. Evaluating the K20 model across lung, colorectal and pancreatic cancers within the TCGA datasets revealed several specific molecular subsets that have increased KRAS dependency, such as the progenitor and squamous clusters, as well as copy number-high pancreatic cancers [31]. In colorectal cancer, our model predicted that CMS4 tumors were not dependent on KRAS. These tumors are characterized by stromal invasion, upregulation of genes involved in epithelial mesenchymal transition, and activation of transforming growth factor β activation which can lead to poor prognosis [22]. Previous studies have shown that these tumors are resistant to MEK inhibitor treatment and may respond to combination therapy with SRC inhibitors [56]. In contrast, the CMS3 KRAS-mutant tumors, which have been previously found to have high KRAS expression, are more dependent on the oncogene [22]. While not much is known about the invasive gene expression subtype in colorectal cancer, clinical studies of the CpG Island Methylator Phenotype-Low (CIMP-L) subtype show that patients with this gene expression profile exhibit more KRAS mutations and show better prognosis than patients with the CIMP-high (CIMP-H) subtype [57,58]. The CIMP-L subtype consists of tumors with low DNA hypermethylation and CpG islands in unique genomic sites that typically lead to metabolic dysregulation. Their dependence on KRAS may be linked to their metabolic demands and may be exploited with MAPK-targeting therapeutics, including KRAS-specific inhibitors. Interestingly, we found that STK11 mutation status in lung cancer did not correlate with increased dependency on KRAS. Previous studies have indicated that co-occurring KRAS and STK11 mutations can lead to worse prognosis in NSCLC patients [59], and a recent study using G12C inhibitor sotorasib found that 50% of patients with co-occurring STK11 responded to KRAS inhibition, however those findings were non-significant and hypothesis-forming [60]. Ongoing clinical trials that combine G12C inhibition with immunotherapy may not be effective as a combination therapy in patients with co-occurring STK11 mutations, who are often non-responsive to checkpoint inhibition [61]. In contrast, clinical studies have not shown a significant difference in response rates to KRAS G12C inhibitors with co-occurring p53 mutations, consistent with our K20 model prediction. Given the recent clinical success of KRAS G12C inhibitors, and with broader classes of KRAS inhibitors on the horizon, it is imperative that simple, yet robust predictive biomarker models be developed that can best define the patients who are most likely to benefit from these novel compounds. Notably, direct KRAS G12C inhibitors have only gained approval in lung cancer in the second-line setting, in part due to the modest responses and duration of response achieved. Hopefully, better predictive models of KRAS dependency, beyond the presence of a KRAS mutation, will enable selection of patients most likely to benefit from these new classes of molecules early on in their care.

## Materials/Methods

### Cell line data

DEMETER2 version 5 data was downloaded from DEMETER2 figshare website (https://figshare.com/articles/dataset/DEMETER2_data/). Only cell lines/disease in the following solid tissues were included: "breast", "colorectal", "esophagus", "gastric", "kidney", "liver", "lung", "ovary", "pancreas", "prostate", "skin", "uterus". Activating mutations in these cell lines include the following: KRAS—A146A, A146T, R164, T74P and any mutations at G12, G13, Q61; BRAF—L597R, V226M, G466V, R682Q, G464E, N581Y, G464V, G469A, L597V, G596R and

any mutations at V600; NRAS—A146V, E132K and any mutations at G12, G13, Q61; HRAS—T58T and any mutations at G12, G13, Q61.

K-mean clustering (k = 3) was applied to K20 dependency scores to determine the KRAS dependency group of the cell lines, like described in E. McDonald et. al [25]. K20 dependency score range of"sensitive", "intermediate", and "refractory" groups were: -2.3 to -1.2, -1.1 to -0.4 and -0.4 to 0.3 respectively. CCLE mutation data was downloaded from CCLE website (version 18Q1).

## TCGA data

TCGA gene expression (RSEM_normalized_log2) for PANCAN was downloaded from https://gdc.cancer.gov/about-data/publications/pancanatlas. The "EBPlusPlusAdjustPAN-CAN_ IlluminaHiSeq_RNASeqV2.geneExp.tsv" file was used. The "mc3.v0.2.8.PUBLIC.maf" file was used to extract KRAS mutation status. We used tumor samples with both mutation and expression profiles for downstream analysis. Only the 14 cancer types used in K20 model training were included in the analysis, which includes KIRC, KIRP, SKCM, LIHC, PRAD, OV, BRCA, ESCA, UCEC, LUAD, STAD, COAD, READ, and PAAD. Genomic subtype data for LUAD [29], COAD+READ [22,30], and PDAC [31] were taken from the respective TCGA annotations.

## Apply K20 model to TCGA

Because there is a difference in gene expression between CCLE cancer cell lines and TCGA human tissues, we rescaled the K20 prediction scores of TCGA-PANCAN samples to align with CCLE cell lines so that the adjusted TCGA-PANCAN prediction scores have the same minimal and maximal values as CCLE prediction scores (formula shown below). The rescaling was done using the 14 TCGA-PANCAN diseases that were used in the K20 model building.

$$K20_i^{\text{TCGA-adjusted}} = \left(K20_i^{\text{TCGA}} - \min_{i=1-n}(K20_i^{\text{TCGA}})\right) \times C + \min_{i=1-m}(K20_i^{\text{CCLE}})$$

$$, where\ C = \frac{\max_{i=1-m}(K20_i^{\text{CCLE}}) - \min_{i=1-m}(K20_i^{\text{CCLE}})}{\max_{i=1-n}(K20_i^{\text{TCGA}}) - \min_{i=1-n}(K20_i^{\text{TCGA}})}$$

## K20 model building

Cell lines were divided into training (67%) and testing (33%) sets balancing for the tissue types and dependency groups. The training set was further stratified into 100 iterations of Monte-Carlo cross validation (MCCV) with 67% training and 33% testing to select the best tuning parameters, α and λ. The α parameter controls the percentage of ridge (L1 regularization) and lasso (L2 regularization). α is the percentage of ridge regularization, so when α is zero, it does only ridge regression, and when α is 1, it does lasso regression. λ is the shrinkage parameter. The larger the λ is, the more shrinkage it has on the coefficients. Binomial models (refractory vs. non-refractory) were fitted to the MCCV training set using Elastic-Net (R package: glmnet). Five different feature sets were evaluated: [1] gene expression of genes with standard deviation (SD) larger than 0.5 in TCGA-LUAD, COAD and PDAC, [2] gene expression from [1] and mutation status of all genes, [3] gene expression from [1] and mutation status of KRAS, [4] gene expression from [1] and mutation type of KRAS, and [5] gene expression from [1] and interaction with KRAS mutation status, where expression value is multiplied to the KRAS mutation status (0 for wild type and 1 for mutant). Elastic-Net parameters were selected to maximize the AUC evaluated in the MCCV test sets. Parameters with the best performance were then used to fit our final model against the complete training set. In the final model, a

total of 20 features were selected, including KRAS mutation status. Selected features and their coefficients are listed in S1 Table. Prediction score is the dot product of the gene expression value and their coefficients.

The final model was then applied to all 444 cell lines and TCGA-LUAD, CORE and PDAC datasets. Heatmaps showing the model gene expression and mutation status were plotted using R package ComplexHeatmap. K20 prediction scores were divided into 3 groups via k-mean clustering (k = 3). Predicted K20 scores for the three TCGA cancer types were also plotted as violin plots by their genomic subtypes.

## TCGA and CCLE Joint statistic calculation

TCGA and CCLE datasets were integrated using the joint dimension reduction method, AJIVE [41]. AJIVE decomposes each input dataset into joint **J** and individual **I** matrices containing shared- or independently- behaving variation, respectively. A joint statistic representing the "joint behavior" for each gene g within a dataset can then be calculated as the log-ratio of the gene's variance $\sigma^2$ between joint and individual matrices (formula shown below).

$\log(\frac{\max(\sigma^2_{J_g},s)}{\max(\sigma^2_{I_g},s)})$, where $\sigma^2_{J_g}$ is variance of gene in joint, $\sigma^2_{I_g}$ is variance of gene in individual, and s is shrinkage threshold.

In this analysis, a shrinkage threshold s of 0.5 was used to set a minimum variance threshold for each gene. This prevents potential artifacts from low variance genes. A "CCLE joint statistic" was calculated for each gene using the decomposed CCLE matrices with higher values representing improved translation from cell line to TCGA data. Thus, we assessed the significance of the K20 CCLE Joint Statistics using the permutation approach proposed by Tusher, Tibshirani and Chu [62] to calculate the false discovery rate (FDR).

## Cells and culture conditions

Cell lines were obtained from the ATCC and tested for Mycoplasma. MIA PaCa-2 (pancreatic carcinoma) cells were grown in Dulbecco's Modified Eagle's Medium (DMEM) media (Gibco) with 10% Fetal Bovine Serum (FBS) (Avantor) and 1% Penicillin–Streptomycin (P-S) (Sigma) antibiotic. SK-CO-1 (colorectal adenocarcinoma) and A-427 (lung carcinoma) cells were grown in Modified Eagle's Medium (MEM) media (Gibco) with 10% FBS and 1% P-S antibiotic. H727 (lung non-small cell carcinoma), H441 (lung adenocarcinoma), H1355 (lung adenocarcinoma), H2030 (lung adenocarcinoma), and KE39 (gastric cancer, kindly provided by the Bass lab) cells were grown in RPMI-1640 media (Gibco) with 10% FBS and 1% P-S antibiotic. HCT116 (colorectal carcinoma) were grown in McCoy's 5A medium (Gibco) with 10% FBS and 1% P-S antibiotic. All cell lines were grown in T75 flasks (Genessee Scientific) at 37°C with 5% $CO_2$/95% air.

## siRNA transfections

All siRNA transfection experiments were completed using Lipofectamine RNAiMAX (Life Technologies) following manufacturer instructions. The sequences of the siRNA oligos are below and as previously described [39].

Negative control: Sense-UUCUCCGAACGUGUCACGUdTdT
Anti-sense-ACGUGACACGUUCGGAGAAdTdT
KRAS sequence #1: Sense-GUCUCUUGGAUAUUCUCGA,
Anti-sense-UCGAGAAUAUCCAAGAGAC
KRAS sequence #2: Sense-CAGCUAAUUCAGAAUCAUU,
Anti-sense-AAUGAUUCUGAAUUAGCUG

### In vitro validation studies

Cell viability in response to KRAS siRNA treatment was evaluated with the CellTiter-Glo 2.0 Cell Viability Assay, which quantifies ATP, using the manufacturer's protocol (Promega). Following trypsinization, cells were resuspended in antibiotic-free culture media and 150uls were added to opaque, flat bottom white 96-well plates (PerkinElmer). Cells were counted on a hemocytometer using a Trypan Blue dye (Sigma) and seeded as follows: MIA PaCa-2 cells were seeded at 2,000 cells/well, H1355, H2030 and KE39 cells were seeded at 4,000 cells/well, A-427 cells were seeded at 5,000 cells/well, and SK-CO-1 cells were seeded at 6,000 cells/well. The KRAS siRNAs were suspended in serum-free media with Lipofectamine RNAiMAX and were reverse transfected at 50uls per well in triplicate starting at 20nM and progressing through a 10-point serial dilution until a final dose of 0.04nM. Plates were sealed with Breathe-Easier sealing films (Electron Microscopy Sciences) to minimize edge evaporation and incubated in culture conditions for six days. The endpoint was experimentally determined as the optimal timepoint for producing robust curves consistently across all cell lines and is consistent with experimental conditions in the field. For viability readouts, 100 uls of media was removed from each well and an equal volume of CellTiter Glo 2.0 (CTG) Reagent was added. Luminescence was measured at 530 nm excitation and 590 nm emission on a Synergy2 fluorescent plate reader (BioTek). Data was analyzed in GraphPad Prism. A negative control siRNA was tested on each cell line and showed no effects on KRAS knockdown or cell viability, therefore we excluded it from figures.

## Supporting information

**S1 Fig. DEMETER2 KRAS dependency score distribution and cell line disease types with KRAS mutations.** (A) Density plot of KRAS dependency score in all 712 cancer cell lines (black line) and in 126 KRAS-mutant lines (red line), p-value = $6.71e^{-30}$ (Wilcoxon rank sum test). (B) Waterfall plot of KRAS dependency scores, color-coded by KRAS mutation status. Red and black color bars represent KRAS-mutant and wild-type cells, respectively. (C) Waterfall plot of KRAS dependency scores, color-coded by KRAS mutation k-mean cluster (k = 3). Red, purple and blue color bars represent sensitive, intermediate, and refractory clusters, respectively. (D) Waterfall plot of KRAS dependency scores in KRAS-mutant cells only. (E) Bar plot of disease types in 126 KRAS-mutant cell lines.
(TIF)

**S2 Fig. ElasticNet model parameter tuning and feature set selection.** (A) Performance comparisons in the training set among the five different feature sets and in different alpha (0.1 to 1) were shown in each plot for AUC versus the number of features. Different colors of lines represent the five different feature sets: gene expression only in black, gene expression and mutation in green, gene expression and KRAS mutation status in red, gene expression and KRAS mutation type in orange and gene expression, KRAS mutation and interaction term between them in blue. The black dotted line is at the highest AUC = 0.94 for easy comparison among plots. (B) Boxplot of AUC comparison at alpha = 0.9 and number of features between 15 to 25 during the MCCV. The black dotted line is at the highest AUC = 0.94. (C) Receiver operating characteristic (ROC) curve comparing the model performance between the K20 model (in red) and the KRAS mutation only model (in blue) in the validation set.
(TIF)

**S3 Fig. KRAS expression in CCLE cell lines and K20 prediction scores across numerous cancer types from CCLE.** (A) Relative KRAS expression by RT-qPCR in HCT116 cells following treatment with the control siRNA, KRAS siRNA #1, and KRAS siRNA #2 at 20nM for 24

and 48 hrs. Error bars indicate SEM. (B) Scatter plot of K20 prediction scores (in x-axis) vs. GI50 scores of the two KRAS siRNAs treated in nine different cancer cell lines (in y-axis) shows correlation (Spearman r = 0.57, p = 0.02). (C) Box plot of KRAS expression by KRAS dependency class and mutation status. Wilcoxon p-values were shown for two group comparisons. (D) Box plot of K20 prediction scores in 12 CCLE cancer types overlapped with the 14 TCGA solid tumors, separated by disease type and KRAS mutation status. Each sample is represented as a dot.
(TIF)

**S4 Fig. Comparison of observed CCLE and TCGA joint statistics.** The observed TCGA and CCLE joint statistics of all genes are shown along the x- and y-axes, respectively. The 19 genes included in the K20 model are shown in blue, while the rest of the genes are shown in grey. The FDR threshold was set to 5% for CCLE joint statistic, which was equal to 0.106 here (shown as a red dotted line).
(TIF)

**S5 Fig. Apply K20 predictive model to TCGA colon and rectum cancer (CORE) patients.** K20 model predicts that most of the colorectal cancer patients with KRAS mutation are KRAS-sensitive with some exceptions and the model prediction scores are associated with several TCGA expression subtypes. (A) Heatmap showing gene expression of 19 classifier genes and KRAS mutation status and types in TCGA colorectal cancer patients. Samples were sorted by prediction scores (left to right: lowest to highest). (B-E) Violin plots of K20 model prediction scores by (B) expression subtypes (KRAS-wildtype (WT): invasive vs CIN, p = 0.0228; invasive vs MSI/CIMP, p = 0.0097; KRAS-mutant (MUT): invasive vs CIN, p = 0.0022), (C) MSI status, (D) methylation status, (E) hypermutation status, and (F) CMS status (KRAS-wildtype (WT): CMS1 vs CMS4, p<0.0001; CMS2 vs CMS4, p<0.0001; CMS3 vs CMS4, p = 0.0004; KRAS-mutant (MUT): CMS2 vs CMS4, p = 0.0262; CMS3 vs CMS4, p = 0.0003; CMS2 vs CMS3, p = 0.0062). Wilcoxon rank-sum test was used for comparisons. (***) = p<0.001, (**) = p<0.01, (*) = p<0.05
(TIF)

**S6 Fig. Apply K20 predictive model to TCGA pancreatic ductal adenocarcinoma (PDAC) patients.** Violin plots of K20 prediction score split into KRAS-wildtype (WT) and KRAS-mutant (MUT) groups by (A) Bailey cluster (MUT: progenitor vs ADEX, p<0.0001; squamous vs ADEX, p = 0.0046, progenitor vs immunogenic, p<0.0001; squamous vs immunogenic, p = 0.0003; squamous vs progenitor, p = 0.0107) and (B) copy number cluster (MUT: low vs non-aneuploid, p<0.0001; high vs non-aneuploid, p<0.0001; low vs high, p = 0.0051). Additional violin plots of K20 prediction scores by (C) lncRNA cluster, (D) Moffitt cluster, (E) methylation cluster, (F) RPPA, and (G) miRNA cluster. Wilcoxon rank-sum test was used for comparisons. (****) = p<0.0001, (**) = p<0.01
(TIF)

**S7 Fig. Apply K20 predictive model to TCGA lung adenocarcinoma cancer (LUAD) patients.** Additional violin plots of K20 prediction scores split into KRAS-wildtype (WT) and KRAS-mutant (MUT) groups by (A) STK11 mutation status, (B) p53 mutation status, and (C) transversion status. Wilcoxon rank-sum test was used for comparisons.
(TIF)

**S1 Table. Selected gene features and coefficients.**
(XLSX)

## Acknowledgments

The authors acknowledge members of the Pecot lab for helpful discussions and feedback. The authors also acknowledge the support from the Bioinformatics Shared Resource at UNC Lineberger Comprehensive Cancer Center.

## Author Contributions

**Conceptualization:** Joel S. Parker, Chad V. Pecot.

**Data curation:** Yihsuan S. Tsai, Joel S. Parker.

**Formal analysis:** Yihsuan S. Tsai.

**Funding acquisition:** Chad V. Pecot.

**Investigation:** Yihsuan S. Tsai, Yogitha S. Chareddy.

**Methodology:** Yihsuan S. Tsai, Yogitha S. Chareddy, Chad V. Pecot.

**Resources:** Chad V. Pecot.

**Supervision:** Chad V. Pecot.

**Validation:** Yihsuan S. Tsai, Yogitha S. Chareddy, Brandon A. Price.

**Visualization:** Yihsuan S. Tsai.

**Writing – original draft:** Yihsuan S. Tsai, Yogitha S. Chareddy, Chad V. Pecot.

**Writing – review & editing:** Yihsuan S. Tsai, Yogitha S. Chareddy, Brandon A. Price, Joel S. Parker, Chad V. Pecot.

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
