## [Decision Letter · Decision Letter 0]

3 Oct 2022

Dear Dr. Pecot,

Thank you very much for submitting your manuscript "An Integrated Model for Predicting KRAS Dependency" for consideration at PLOS Computational Biology.

As with all papers reviewed by the journal, your manuscript was reviewed by members of the editorial board and by several independent reviewers. In light of the reviews (below this email), you are welcome to submit  a significantly-revised version that takes into account the reviewers' comments in a detailed manner.

We cannot make any decision about publication until we have seen the revised manuscript and your response to the reviewers' comments. Your revised manuscript will be sent to reviewers for further evaluation.

Sincerely,

James Gallo

Academic Editor

PLOS Computational Biology

Sushmita Roy

Section Editor

PLOS Computational Biology

Reviewer's Responses to Questions

**Comments to the Authors:**

Reviewer #1: The manuscript by Tsai et al presents a proposal for a prognostic panel of 20 markers (K20) for sensitivity to KRas modulation across multiple cancer types. KRas mutation is surely significant across human cancer and although recently druggable, still present major clinical challenges. There is precedent for similar types of marker panels such as the PAM50 set for breast cancer. The authors integrate large publicly available datasets in potentially new and what seems to be interesting ways to arrive at their K20 panel that is claimed to be predictive of KRas dependency. Some cell line data, both from the publicly available sources and their own independently generated data, are presented to support this claim. Some of the data could be presented in clearer ways to provide a convincing demonstration. A major concern is that it seems KRas mutation status and its expression level form the predominant basis of the panel…are the other 18 markers really needed? The mutation alone gives AUC of 0.854, whereas adding the 19 expression levels gives 0.954, so the improvement may be seen as modest although maybe important. Major decisions are also made in a rather ad hoc manner, such as the stratification into three KRas dependency groups although the data form a continuum---this is not necessarily a problem but the reader is left wanting for more discussion and justification. Also other feature sets with better AUC are left behind. The authors apply the K20 panel to TCGA data. While application to clinical data from TCGA on its surface seems useful, this Reviewer is unsure how much scientifically is truly added with these applications, since there seems to be no KRas dependency data in TCGA that could be used to test the K20 model predictions. TCGA analysis takes a substantial part of the manuscript but seems mostly speculative. Thus, overall while this paper has some potentially interesting results, I am not sure how impactful this panel truly is. Some additional specific points are listed below.

1. Introduction, page 3, 2nd paragraph. The statement is made that attempts to block downstream MAPK signaling has not yet proven to be clinically effective. However what about MEK and Raf inhibitors? This section needs more nuance and balance.

2. The methods section in general would benefit from far more detail. For example, details of how CCLE data was “aligned” with TCGA data seem critical but it is not clear nor reproducible from what is written. In the section on K20 model building, what is interaction with KRAS mutation status? For the CCLE and TCGA joint statistic calculation, why is there a max variance (how are there several?)? Treatment of cells for siRNA experiments is also lacking detail, such as when siRNA was added, how long cells were given to attach, what kind of plates were used, and so on. These are just some examples that arise from a lack of sufficient detail in the methods.

3. Results page 12, what is alpha and why is it significant that it takes different values with different feature sets? How was binary classification accomplished with the expression levels?

4. The authors discuss that KRas expression levels are by far the highest contributor to the K20 panel outside of mutation status (Fig. 1C). Are the other 18 expression levels even needed?

5. Fig. 2D is the main experimental validation summary but it is unclear how these siRNA dose response experiments support the claims that the K20 panel has predictive power for KRas dependency. Maybe a plot of sensitivity vs. K20 score like 2E could demonstrate this better.

Reviewer #2: Tsai and colleagues seek to predict response to KRAS G12C inhibitors by generating a score (K20) from some 400 cell lines in DEMETER2 and validating it using TCGA and other data. The manuscript needs minor language revision.

The K20 model embeds several genes that are known in the field and by orthogonal approaches to be linked to KRAS signaling, which appears highly reasonable. A suggested improvement would be to generate the K20s for one tumor type at a time for colorectal, lung and pancreas cancers in the discovery part of the project, as that would potentially reduce background from analyzing different tumor types in the same score and result in a better performing score for each tumor type. The text relating to figs 3-5 feels repetitive/cataloguing and should be condensed into a single display item/figure panel with more concise text.

Reviewer #3: In their manuscript entitled “An Integrated Model for Predicting KRAS Dependency”, the authors developed a model to predict KRAS dependency, and externally validated the performance of the model using genetic depletion assays and an external datasets. In addition, they applied the model to TCGA pan-cancer datasets. This study provides a model by integrating expression, mutation data and RNAi screening data to predict KRAS dependency, for improving the stratification of KRAS mutant tumors to benefit the therapies of KRAS inhibitions. This work is valuable from the perspective of the precision medicine of KRAS mutant tumors. Here I would like to bring out the following points from this work:

1. To help the readers better understand this study, the authors are suggested to explain the term “kras dependency score” where it first appears in the manuscript.

2. There are several similar terms in the manuscript, such as k20 predictive score, K20 prediction score, and predicted k20 score. If these terms have the same concept, they are suggested to have a consistent naming. Also, explain this term in the first place where it appears. In my understanding, k20 prediction score is the predictive dependency score in the model. In this way, when the authors used a cutoff to define the cell lines to be “sensitive” or “refractory”, how to use a reasonable cutoff? In external validation of the model, they used 1.477 as a cutoff. However, no information about the cutoff was given in the training model. Is the cutoff 1.477 consistent with the training model? Moreover, is this value reasonable to distinguish “sensitive” from “refractory” from the perspective of biological meaning?

3. The authors used five different features to develop the model. I don’t well understand the feature set “the interaction between KRAS mutation status”. This feature set can be described with more details.

4. In page 12 line 264, gene expression index was calculated by reads per kilobase normalized on log2 scale. I’ m wondering if gene expression index was measured by FPKM, TPM or others in the manuscript? Was gene expression also normalized by sequencing depth?

5. The Fig. 2A and 2B actually don’t provide the information of “sensitivity of 0.95 and specificity of 0.89” which is described in the manuscript. In Fig. 2C, though the model prediction scores are highly correlated with the KRAS dependency scores, the prediction scores in y-axis actually can’t distinguish “sensitive”, “intermediate” and “refractory” very well. The authors need to well address this point and clearly show that their model is potentially clinically effective.

6. In the model, KRAS mutation and KRAS expression have the most contribution on KRAS dependency score. Beyond these well-known biomarkers, more efforts should be made to investigate the other features in the model to understand the KRAS independent tumors for developing the new strategies of precision medicine.

7. In Fig. 1A, the information of “kras mutant” and “sensitive” is shown in the same red color. The different colors can be used in the same figure panel. In addition, red and green colors in the same panel can be replaced with other colors. In Fig. 1C, add the name of x-axis. In Fig. 1D, what does the number after the gene name represent for? This could be added in the figure legend.

Reviewer #4: Tsai, et al, describe a model based on gene expression and mutational status to predict KRAS dependency of tumors. The model is trained on cell lines and validated on TCGA and in vitro dose response experiments.

General comments:

The concept of modeling RAS dependence using gene expression data is not a new one: see (Singh, Cancer Cell, 2009; Way, Cell Reports, 2018; Guinney, Clinical Cancer Research, 2013). Authors need to clearly articulate the advantage/comparison of their approach over prior models.

KRAS activating mutations are not defined. Is any amino acid concerned considered, or only those with canonical codon changes?

How do the authors justify the exclusion of NRAS / HRAS from their model?

Fig 1A needs to overlay other activating RAS mutations or those that may phenocopy activation of this pathway (NRAS, HRAS, BRAF). This applies to Fig 3A as well, which by focusing exclusively on KRAS may be missing activated samples from NRAS and/or BRAF, esp in GI cancers but others as well. In Line 340/341, the authors attribute this to false negative KRAS mutations, but is more likely to be explained by alternative RAS activating mutations.

Specific comments:

Page 5-6: “we rescaled the K20 prediction scores of TCGA-PANCAN samples to align with CCLE cell lines so that the adjusted TCGA-PANCAN prediction scores have the same minimal and maximal values as CCLE prediction scores.”

This is an important point that seems to be glossed over, I would like some more detail on the approach and justification for this. Aligning the max and min could be sensitive to outliers, and I’m also worried about tissue specific effects and the two cohorts not being matched in proportion of samples from each tissue type.

Page 6: ‘gene expression of genes with standard deviation (SD) large than 0.5’ large -> larger

Page 10: In Fig 1B the AUC of the continuous output of the Mut+Exp Model is compared to the AUC of a binary Mut Model. This is misleading because the AUC of a binary model will very often be lower because the ROC curve only has 3 points with the curve interpolated between them. It would be most convincing to compare binary classification scores between the models using the threshold of the Mut+Exp Model selected for external validation

Page 12: “we only included genes that had variations (SD>0.5 in log2 normalized gene expression) in the lung adenocarcinoma (LUAD), pancreatic adenocarcinoma (PDAC), and colorectal cancer (CORE)”

How many genes was this? I’d be curious to know the number of possible features of all the compared models.

Page 12: “We selected the feature set with “gene expression and KRAS mutation status” as our final model because of the minimal performance difference (i.e., AUC=0.94 vs. 0.93) and the smaller number of features required”

I agree with the parsimonious approach of selecting a simpler model with similar performance. The justification here and in Fig S2A, however, is a bit weak. This is partially due to Fig S2A being difficult to read. Showing the standard error of the AUC from the cross validation could strengthen this by showing that the simpler model does not have significantly worse performance. I’d be curious to see if the selected model is the simplest model within 1SE of the highest AUC as opposed to a model of just KRAS mutation and KRAS gene expression for example.

Page 12: “Unexpectedly, aside from KRAS mutational status, we found that KRAS expression itself played the strongest role in predicting KRAS dependency (Fig. 1C)”

- I don’t think this is unexpected.

Page 15: I find Fig 2D hard to read, it is difficult to tell the difference between the circles and squares when the error bars are small

Page 16: “many KRAS wild-type GI tract cancer patients also had low K20 scores, which we suspect may be due to false negatives in KRAS mutation detection”

I think the authors are referring to TCGA patients here but based on Fig3B, the K20 scores are lower for KRAS wt cell lines from GI cancers as well. Could this be explained by tissue specific expression differences or copy number changes?

Page 18: I’m curious why the copy number analysis was left out of the CORE TCGA analysis.

Page 18: Authors should use the more widely accepted Consensus Molecular Subtypes (see Guinney, Nat Med, 2015) to analyze CORE samples. Moreover, line 381-387 are not results, and speculation on the part of the authors. Move to discussion.

**Have the authors made all data and (if applicable) computational code underlying the findings in their manuscript fully available?**

Reviewer #1: **No: **I did not see any mention of the code being available (just public parts of the data sets).

Reviewer #2: Yes

Reviewer #3: Yes

Reviewer #4: Yes

PLOS authors have the option to publish the peer review history of their article (what does this mean?). If published, this will include your full peer review and any attached files.

Reviewer #1: No

Reviewer #2: No

Reviewer #3: No

Reviewer #4: No
---

## [Decision Letter · Decision Letter 1]

5 Feb 2023

Dear Dr. Pecot,

Thank you very much for submitting your manuscript "An Integrated Model for Predicting KRAS Dependency" for consideration at PLOS Computational Biology.

As with all papers reviewed by the journal, your manuscript was reviewed by members of the editorial board and by several independent reviewers. In light of the reviews (below this email), we would like to invite the resubmission of a significantly-revised version that takes into account the reviewers' comments. At this stage, most of the concerns are confined to a single reviewer.

We cannot make any decision about publication until we have seen the revised manuscript and your response to the reviewers' comments. Your revised manuscript is also likely to be sent to reviewers for further evaluation.

Sincerely,

James Gallo

Academic Editor

PLOS Computational Biology

Sushmita Roy

Section Editor

PLOS Computational Biology

Reviewer's Responses to Questions

**Comments to the Authors:**

Reviewer #1: This is a revised version of a manuscript by Pecot and others that focuses on developing a predictive signature for KRas modulation dependency across carcinomas. The motivation that biomarkers in addition to genetic status only for predicting KRas inhibition response remains well received. The authors have added to the manuscript to clarify previously unclear aspects.

The major concerns I retain relate to what I believe is the crux of the paper in the independent experimental validation in Fig 2 and S3.

1. The authors select six cell lines, two from each of the sensitive, intermediate, and refractory categories, with one being KRas WT (intermediate), and perform KRas siRNA dose responses and assay cell metabolic output (CellTiterGlo). On a positive note, the model seems to have correctly predicted the sensitivity of the KRas WT line.

a. I did not see any confirmatory experiments that KRas was indeed knocked down, or description of controls with non-targeting siRNA.

b. It is reported that the cells were allowed to grow in 96 well plates for six days following seeding / reverse transfection. This is quite long compared to typical dose response experiments, at least for small molecules (2-3 days), so some justification is warranted here.

c. The authors have added Fig. S3A to better show a potential trend between K20 prediction score and response to KRas siRNA (GI50). Although a linear trend line is shown with r=0.62, the trend seems completely dependent on one cell line with a highly negative K20 prediction score (see also d below). Besides this data point, the trend is that there is essentially no dependence between the K20 prediction score and the siRNA GI50.

d. Although which cell line referred to in point c above does not manifest from the plot, it may be SKCO1, from the sensitive cluster, based on the highly negative K20 Prediction score. This looks to be ~-1.6 from the plot, but is listed as -1.95 on Fig. 2D legend. Along these lines, the values for K20 scores in the Fig 2D legend do not seem to match those in S3A—reconciliation is needed.

2. The authors apply the K20 panel to an independent dataset on lung cancer cell lines with KRas mutation treated with a mutant KRas inhibitor. Again, the preponderance of data seems to indicate most cell lines are resistant to the drug, yet have a wide range of K20 prediction scores between -1 and 0, which seems to indicate sensitivity to KRas inhibition based on the authors cutoff of ~1.4. How robust is the result to the lone data point on the far left?

Other comments:

1. Fig S3 has two panel Bs.

2. It is still logically unclear to me why to make three clusters of cell lines when a binary outcome is modeled.

3. Page 7, line 131, it is confusing to introduce the K20 score and 19 genes before development of the model (I am not sure if that’s what the authors intended but it confused me).

4. The authors address prior modeling work on KRas sensitivity gene signatures (a point raised by Reviewer 4) by saying a pan-cancer signature is needed, which has not been done. However it is not clear why cancer-type-specific models would be suboptimal. In fact, they may be better at predicting due to tissue type differences across cancer types. In their response to Reviewer 2, the authors say there simply was not enough data yet to build robust classifiers in a cancer specific way, which seems contradictory here.

5. The authors select genes that had variations (SD > 0.5) in lung, pancreatic and colorectal cancer mRNA datasets in TCGA. Variations with respect to what? This was not clear from the methods.

6. It was not described in the methods how the K20 score was actually calculated from the 20 features. Is it a dot product of the feature value vector and their coefficients?

7. The pages and line numbers referenced in the rebuttal letter do not line up with the revised paper.

Reviewer #3: Regarding the 5th comment, gene expression index was just normalized by sequencing depth, it should be RPM. RPKM is the normalized expression index with sequencing depth and gene size.

Reviewer #4: The authors have addressed all issues to my satisfaction.

**Have the authors made all data and (if applicable) computational code underlying the findings in their manuscript fully available?**

Reviewer #1: **No: **I did not see code or input data to replicate results or use the K20 model

Reviewer #3: Yes

Reviewer #4: Yes

PLOS authors have the option to publish the peer review history of their article (what does this mean?). If published, this will include your full peer review and any attached files.

Reviewer #1: No

Reviewer #3: No

Reviewer #4: No
---

## [Decision Letter · Decision Letter 2]

10 Apr 2023

Dear Dr. Pecot,

We are pleased to inform you that your manuscript 'An Integrated Model for Predicting KRAS Dependency' has been provisionally accepted for publication in PLOS Computational Biology.

Best regards,

James Gallo

Academic Editor

PLOS Computational Biology

Sushmita Roy

Section Editor

PLOS Computational Biology

Reviewer's Responses to Questions

**Comments to the Authors:**

Reviewer #1: The authors have addressed all remaining comments.

**Have the authors made all data and (if applicable) computational code underlying the findings in their manuscript fully available?**

Reviewer #1: Yes

PLOS authors have the option to publish the peer review history of their article (what does this mean?). If published, this will include your full peer review and any attached files.

Reviewer #1: No

---

## [Editor Report · Acceptance letter]

25 Apr 2023

PCOMPBIOL-D-22-01086R2 

An Integrated Model for Predicting KRAS Dependency

Dear Dr Pecot,

I am pleased to inform you that your manuscript has been formally accepted for publication in PLOS Computational Biology. Your manuscript is now with our production department and you will be notified of the publication date in due course.

With kind regards,

Zsofia Freund
